# Nurse Who Had MERS-CoV Complications with A Near-Death Experience during Pregnancy: A Narrative Analysis

**DOI:** 10.3390/healthcare12030298

**Published:** 2024-01-24

**Authors:** Abbas Al Mutair, Zainab Ambani, Alexander Woodman, Chandni Saha, Hanan F. Alharbi, Alya Elgamri

**Affiliations:** 1Research Center, Almoosa Specialist Hospital, P.O. Box 5098, Al-Ahsa 36342, Saudi Arabia; abbas.almutair@almoosahospital.com.sa (A.A.M.); chandni.saha@almoosahospital.com.sa (C.S.); 2School of Nursing, University of Wollongong, Wollongong, NSW 2522, Australia; 3Almoosa College of Health Sciences, P.O. Box 5098, Al-Ahsa 36342, Saudi Arabia; 4Nursing Department, Prince Sultan Military College, P.O. Box 946, Dhahran 34313, Saudi Arabia; 5King Saud bin Abdulaziz for Health Sciences, King Abdullah International Medical Research Center, Ministry of the National Guard Health Affairs, P.O. Box 3660, Al-Hasa 36428, Saudi Arabia; ambaniz@ksau-hs.edu.sa; 6Department of Family Medicine, School of Medicine, Arabian Gulf University, Manama 329, Bahrain; alexwoodman.ucla@gmail.com; 7Maternity and Child Health Nursing Department, College of Nursing, Princess Nourah bint Abdulrahman University, P.O. Box 84428, Riyadh 11671, Saudi Arabia; 8Faculty of Dentistry, University of Khartoum, Khartoum 11111, Sudan; a.elgamri@hotmail.com

**Keywords:** near-death experiences, MERS-CoV complications, pregnancy, psychological experiences, physical complications, Labov’s model of narrative analysis

## Abstract

Background: According to prevailing views in neuroscience, near-death experiences (NDE) occurring after severe head trauma, critical illness, or coma are often life-transforming experiences in which no awareness or sensory experience of any kind is possible. Although there are general patterns, each case is quite different from the other and requires accurate recording and reporting to potentially explain the phenomenon. Aim: This narrative study aimed to explore a pregnant woman’s NDE due to complications from MERS-CoV. Methods: This was a qualitative narrative study with the administration of two unstructured interviews. After the second interview, the participant completed the Greyson NDE scale, presented through descriptive statistics. Qualitative data were analyzed using Labov’s model of narrative analysis through abstract, orientation, complicating action, evaluation, resolution, and coda. Results: The Greyson scale resulted in a total score of 12, confirming that the patient had experienced an NDE. Labov’s model of narrative analysis revealed that the patient’s experience was not limited to the NDE but had implications for her recovery and life. The patient experienced all three types of NDEs: out-of-body, transcendental, including the transition of consciousness to another dimension, and a combined experience. She also suffered from prolonged hallucinations, neuropathy, and post-intensive care syndrome (PICS). At the same time, the patient experienced what is known as NDE aftereffects, which are caused by a change in beliefs and values; she began to lead a more altruistic life and became interested in the meaning of life. Conclusions: NDE survivors should be encouraged to talk more and share their stories with others if they wish. This study not only investigates the NDE but also considerably adds to the existing literature by integrating a unique cultural view from a country outside of the US and other Western nations, and it highlights the significant role of healthcare providers in NDEs and the importance of communication with comatose patients. It underscores the need for compassion when dealing with patients with NDEs.

## 1. Introduction

Near-death experiences (NDEs), defined as a report of unusual memories associated with a period of unconsciousness during a life-threatening illness, injury, or resuscitation after cardiac or respiratory arrest, have been an area of research in psychology and psychiatry since 1975 [1,2,3]. According to Raymond A. Moody, Jr. (1975), patients are considered “near death” if they are physically weakened to the point that they will die irreversibly unless their condition improves. In an NDE, the person appears to be awake and watches their body and the environment from a location outside of their physical body [2,3].

NDEs have different interpretations from different perspectives, including religious, spiritual, philosophical, and scientific [4,5]. Each view has its perspective and interpretation, but the fact is that this phenomenon is worth further research since it is reported that about 17% of people have experienced NDEs [6]. The prevalence of this phenomenon in patients who have suffered cardiac arrest ranges from 3.6 to 23%. Further data from retrospective studies showed that 43 to 48% of adults and 85% of children suffering from life-threatening illnesses may have experienced an NDE [7].

Examining the perspective of survivors, cumulative evidence suggests that the occurrence of an NDE leads to positive consequences, such as a more charitable life, spiritual growth, a search for the meaning of life, fewer materialistic values, or a decreased fear of death [6,7,8,9,10]. From a scientific point of view, some scientists have interpreted NDEs as delusions that a person experiences during cardiac or brain arrest as a result of certain electrical waves released within the dying brain. Others have argued that these theories are not based on sound scientific foundations and that many observations may refute these theories. Since different studies have reported conflicting results with no consensus on NDEs, more evidence needs to be sought in different populations and diseases or causes to improve the understanding of this phenomenon [5,6,7,8].

Middle East respiratory syndrome coronavirus (MERS-CoV) was first reported in Saudi Arabia in 2012. It was also reported in many neighboring countries [11,12]. In the context of the current study, it is essential to emphasize that the patient’s experience of MERS-CoV resulted in development of her NDE.

MERS-CoV in its initial emergence resulted in multiple sporadic cases in different regions. The causing virus exhibited multiple intrafamilial transmissions and large outbreaks in healthcare settings contributing to clusters of infection. Additionally, the impact of MERS-CoV extended to health care situations, where it triggered large-scale occurrences, underlining the increased threat and encounters correlated with nosocomial infections [13,14].

In health care settings, the transmission of MERS-CoV has been facilitated by overcrowding, poor infection control measures, unrecognized infections, and superspreading phenomena [15]. The Government of Saudi Arabia organized a MERS-CoV awareness campaign led by the Ministry of Health, which involved all primary, secondary and tertiary health care facilities and public health officials. However, the virus continued to spread among health care workers on the front lines of managing and treating patients in the community [11,14,16]. One of the unique cases reported during this outbreak was a Saudi nurse who contracted MERS-CoV at 32 weeks of pregnancy and experienced an NDE [17]. Thus, Alserehi et al. (2016) reported the case of a 33-year-old woman working as a critical care nurse who, despite respiratory failure requiring mechanical ventilation and admission to the intensive care unit (ICU), delivered her baby by emergency cesarean section. The newborn was kept in the neonatal unit for observation and was fed formula instead of breast milk [16]. Repeated PCR analysis of nasopharyngeal swabs showed a consistently negative result for MERS-CoV. However, despite this positive outcome for the baby, the mother’s health was seriously compromised as she suffered from an NDE in addition to MERS-CoV [17]. Years after those physical and emotional ordeals, the mother agreed to participate in this study and present her case as a narrative study.

## 2. Materials and Methods

### 2.1. Study Design

This is a qualitative narrative study of a 33-year-old nurse who experienced MERS-CoV infection and an NDE in Saudi Arabia. First presented at an academic conference in Jeddah, the participant agreed to share her experiences through an in-depth interview. The first interview was conducted in May 2018 and captured the immediate aftermath of the nurse’s encounter with MERS-CoV. This study takes a comprehensive approach by including a follow-up interview conducted after four years in June 2022 to explore the long-term effect of her previous experiences, especially during the COVID-19 pandemic. The aim of the second interview is to delve into the long-term effects of the nurse’s previous experiences, particularly in light of the ongoing global challenges posed by the COVID-19 pandemic. This dual-interview approach allows for an in-depth exploration of the patient’s perspectives, coping mechanisms, and the enduring impact of her encounter with both MERS-CoV and the subsequent NDE over an extended timeframe.

### 2.2. Data Collection

The first phase of the study included two unstructured in-depth online interviews. The first interview lasted about 1.5 h. The second interview lasted about 1 h and 15 min. After the second interview, the participant completed the Greyson NDE scale, a 16-question scale structured around four domains: cognitive, affective, paranormal, and transcendental components of the NDE. Each question out of 16 has 3 options with different scores ranging from 0 to 2. A total score of 7 or higher out of 32 should be interpreted as the person having an NDE. The scale has been found to have good internal consistency, good half–half reliability, and good test–retest reliability [18,19].

### 2.3. Data Analysis

The Greyson scale findings were presented through descriptive statistics. Qualitative data were analyzed using William Labov’s model of narrative analysis, which allowed us to focus on the oral narrative of the participant [20]. According to Labov, observing and recording sound changes is not enough to understand the process of change: one must view these changes in the context of the community in which they occur, as they occur. Thus, given the sensitivity of the topic being studied and the uniqueness of the situation and environment, this approach has been considered the most reliable for analyzing and presenting the data as part of this research [20,21]. This story thoroughly investigates a narrative within the domain of individual interaction, dissecting it across six categories essential for development, presentation, and interpretation. Through a careful analysis, this study seeks to solve their nuanced relationship, shedding light on how the following categories jointly influence the narrative experience: *abstract*, *orientation*, *complicating action*, *evaluation*, *resolution*, and *coda.* As shown in Table 1, the *abstract* encapsulates the reason the story is being told, *orientation* provides details of time and setting, the *complicating action* indicates an event that changed the course of action, *evaluation* refers to the narrator’s attitude about the event, *resolution or result* is the final outcome, and *coda* brings the audience back to the present time [20,21,22].

### 2.4. Reliability and Validity

The reliability and validity of the research were achieved through the following components. Credibility was established through long-term interactions aimed at understanding the participant’s language and perspectives. The results were sent to the participant for comments and possible remarks. Transferability was ensured by a detailed description of data collection and procedures, allowing future researchers to replicate these data if there were similar cases in a different setting. Dependability was ensured by the good documentation of the data analysis process. The researcher stored the raw data, that is, the audio recordings and scale data, for possible audit procedures. Confirmability was established when the collected data were checked and re-checked throughout the data collection and analysis process to ensure that the procedures could be repeated by others. The trustworthiness of the data was confirmed by conducting a follow-up interview after four years, during which it turned out that the participant remembered everything as it was and even used the same wording when describing the same events. Additionally, data triangulation was performed by incorporating a published case report by Alserehi et al. (2016), which included data from the patient, medical staff, and medical records from the emergency and gynecology/obstetrics departments of the hospital, as well as a presentation made in 2018 in Jeddah [17].

### 2.5. Procedures

IRB approval for this study was obtained from Almoosa Specialist Hospital (IRB journal number: ARC-22.07.03). This study was conducted in accordance with the ethical principles of the Declaration of Helsinki and good clinical practice. The participant voluntarily gave full verbal and written consent to this study, and her personal data were published with her consent. Although the participant agreed for her name to be published as well, the decision was made to anonymize her name to avoid any future complications. Therefore, for the purposes of this research, the term “patient” will be used, consistent with the earlier article by Alserehi et al. (2016) [17]. The patient was asked to speak in her preferred language; she mixed Arabic and English in her answers. Both interviews were recorded and then transcribed verbatim. Part of the interview, voiced in Arabic, was translated at the transcript level. The context of the dialogue was preserved by preserving non-lexical utterances.

## 3. Results

### 3.1. Descriptive Data

The patient is a Saudi patient safety nurse who became pregnant with her child after undergoing in vitro fertilization (IVF) in 2013. She was 31-weeks pregnant when the MERS-CoV outbreak occurred. At that time, she was working as a nurse in the medical-surgical intensive care unit (ICU). As part of her duties, she encountered a case of pneumonia that later tested positive for MERS-CoV, thereby infecting the patient. She was treated in the emergency department and then transferred to the ICU, intubated, and sent for an emergency cesarean section at 32 weeks of gestation. The patient was kept under deep anesthesia during treatment for acute respiratory distress syndrome. After 32 days from birth, she had the opportunity to visit her son for the first time in the neonatal ICU (NICU). This was followed by a long physical and psychological recovery.

The result of the Greyson NDE scale filled in by the patient showed a total score of 12. A score of 7 or higher suggests that the person has an NDE (Table 2). Further details of the scale revealed that everything seemed to be happening simultaneously—time stopped or lost all meaning, and her thoughts sped up. She further reported that she felt at one with the world, and that her feelings were very bright and clear and were surrounded by incredible light.

The quantitative data analysis was followed by a narrative analysis presented according to Labov’s structural approach.

### 3.2. Qualitative Data

#### 3.2.1. Abstract—What Was This About?

The patient began her narrative with an abstract description of her experience with MERS-CoV, her symptoms and the fears she experienced due to her long-awaited labor. She then recalled fragmented images due to worsening MERS-CoV symptoms and a subsequent loss of consciousness.

“I started having symptoms, but I did not link it to MERS-CoV. I thought it was just the flu. However, after 7 days, I thought, “this is not the common flu”. Thus, I went to family medicine doctors and explained to them that I am an ICU nurse, and we have a patient that might have MERS-CoV and I am having an IVF baby, so, they did the swab, and it was negative. They cannot prescribe me any medication, because I am pregnant. On that same day, I found out that one of my patients is one of two suspected cases of MERS. Later that night, I was admitted to the emergency room, but I cannot remember this part. They told me that I was there for 2 days and from there I was transferred to OB/GYNE.”

#### 3.2.2. Orientation—Who or What Are Involved in the Story, and When and Where Did It Take Place?

The orientation of the patient’s narrative identifies the time, place and situations followed by a confirmation of her MERS-CoV infection and subsequent actions.

“In the OB/GYNE, I remember a few memories flashes, I was on a non-rebreather mask, and I couldn’t breathe properly. I asked to call the Rapid Response Team and the team from ICU came. The ICU doctor assessed me.”

The patient was recommended to have complete bed rest. She was further told that she could have an emergency cesarean section but would need to be admitted to the ICU. After this, her condition began to rapidly deteriorate. She developed respiratory failure and required mechanical ventilation. Doctors decided to intubate the patient and put her under deep sedation. However, her thoughts were mostly related to all those who might have been affected due to her health condition.

“I remember I was very frustrated; I couldn’t breathe. I was seeking air. I was thinking about what would happen. I was afraid that my mother might have gotten exposed to the virus because my mother and my sister accompanied me to the hospital. I have a husband who is almost immunocompromised. I felt like maybe one of my family would get sick also. I had all these fears plus I was more worried about my baby. Is it going to affect him? Is he going to be alive? I was just 31 weeks at that time. I didn’t know what would happen. Is he going to live? Am I going to see him?”

#### 3.2.3. Complicating Action—Then, What Happened?

After deep sedation and intubation, the patient reported that she did not know what had happened to her. Doctors later informed her that she was taken to the operating room and that by cesarean section she gave birth to a healthy but premature male child weighing about 1.79 kg. The infant was kept in the NICU for over 3 weeks on continuous positive airway pressure (CPAP) for the first five days. He was then fed through a nasogastric tube. After delivery, the patient was transferred to the ICU, still intubated and under deep sedation. It was during this period that the patient reported having dreams and out-of-body experiences.

“I felt constrained in a place, and I am trying to get out of it and trying to search for someone, but I don’t know where. I was trying to search for my son, always searching and searching...”

In her further narrative, the patient described being in a crowded place with many people and someone telling her that she was fine and so was her baby. This dream was later confirmed by medical staff, who said that she not only dreamed, but also heard them talking to her supportively while under deep sedation.

“They told me later that one of the ICU consultants was the one who discovered this. She was the one who came and talked to me while I was intubated, and she said my son is doing very well. He is going to be discharged from the neonatal ICU and be strong so that you can take care of him. At that time, I was having tachycardia, I think 180 bpm or more, hypertensive even, and everything went to normal suddenly. Everyone was surprised, and since then, they have started talking to me every day.”

After this, the patient reported that it became a regular occurrence, with some staff regularly visiting her and talking to her, encouraging her to return back to life. In fact, the patient remembered all the details of the conversations that took place in her room, which were later confirmed by the staff word for word. However, she continued to be in an unusual sensation, incomparable to just a dream, where her feelings changed in a matter of seconds, and she could transform from one dimension to another.

“I remember times when I was sedated, I had dreams that I was set in a place like a desert and in another dream at a white palace, and the whole time I was searching for my son. I am thinking that maybe I was on the other side of life? I don’t know [tears]. These are the things that make me always think that I was very close to death, but Allah gave me a chance to live again.”

After 13 days of sedation, the patient remained in the ICU for an additional two weeks for stabilization. During this time, her condition worsened somewhat. At some points, the medical staff expected her to die and tried several times to extubate her, but without success. Then, a decision was made to conduct extracorporeal membrane oxygenation (ECMO). However, the medical staff decided to give her a few more chances to recover. The patient eventually improved, her oxygen requirement decreased, and she was able to resume spontaneous breathing. As a result, she was taken off the ventilator and transferred to a ward.

“I was happy to see the people I knew despite the fact they were scarily wearing masks as if I was in space. But at the same time, I felt I remembered these people, I knew them. Some of them, I did not recognize, I mixed them up. At the same time, I had angry feelings, I don’t know why. I felt they were giving me medication that made me paralyzed.”

The above narrative describes the patient’s near-death experience and her experience of “life” outside the body. Apparently, she was somewhat conscious, but these were scattered sensations that changed in a matter of seconds, transferring her into an unconscious state filled with fear and the unknown. The patient continued her narrative by sharing what happened after she awoke from sedation.

#### 3.2.4. Resolution—What Finally Happened?

Once the patient was extubated, she began a new part of her journey where she suffered from hallucinations.

“Many times, I saw beside me a man lying on the bed, and I saw my husband outside angry. My husband was talking to the nurses about my health conditions, but I was thinking that he was angry because I was lying beside a man. I remember when the Head Nurse came to me, I told him my husband is angry because Abdulkhaliq [the imaginary man] is lying beside me. He asked me: who was Abdulkhaliq? I said: here, there is a patient beside me. He told me: there is no one beside you!”

Further hallucinations led to the patient feeling out of place as she saw herself leaving the room and seeing things that did not exist in reality.

“I saw a sign hanging outside my room door saying “It is forbidden to visit the patient by order of Dr. So and so”. When my boss came to visit me, I asked him why you put a sign on the door that says no visit? He said, “there is no sign”. Then suddenly my brain understand that it was not real.”

She had a vision that she got up from the room and went to the NICU, took her son, was discharged from there and gave him to her brother.

“I asked him “take him with you, I will stay here long”. I was sure I did this. At that time, my friend’s husband was the doctor who took care of me. I told him I discharged my sone today and I sent him home with my brother.”

Along with the hallucinations, the patient experienced severe neuropathy, muscle atrophy, frustration, and conflict of feelings.

“When they were doing my morning care and positioning, I felt severe pain. I felt as if I fell from the 10th floor because I feel as if all my joints are moving and getting out of their places. I was happy that everyone was around me, all of them were talking to me or trying to help me, but at the same time, there was an angry feeling, what are you doing to me?”

After some time, her hallucinations disappeared, and she began a new part of her journey: meeting her son.

“I felt I wanted to see the baby; I can’t wait to see him. While I was going to see him, I was anxious and afraid. What does he look like? Is his size normal? Is he intubated? I wanted to see him but was afraid at the same time. Then, they took me to the neonate ICU. He was 32 days. He was very little. Thanks to good, he started to drink milk formula normally. I stayed there for about half an hour.”

This tender moment was described further:

“It was a moment, ah, strange feeling that this creature is mine. I was crying when I saw him. It was a strange feeling, I don’t know, I felt he was a gift that God gave me. I wasn’t imagining this was my son, I was looking at him and wondering if he was real? Is he really my son?I saw him when he was 32 days, um, the biggest thing I could do was touch his face, um... I was wearing a mask, I wasn’t able to carry him, Ummm, [tearing] I had feared, especially in the beginning as I have severe muscle waste, Ummm, with physiotherapy, I tried my best to do all kinds of movements because I want to recover quickly because I want to get out of the hospital quickly. I stayed for two and half months in the hospital. Um, I was thinking, if I stay in the house, me and him, how am I going to look after him, I was afraid, I’m his mother Uh, I want to take care of him.”

In the last week of hospitalization, the patient asked the medical staff to allow her son to sleep next to her. She felt that she had to do everything for him, even though she could not carry him at that moment. That was the end of her near-death experiences as a hospitalized patient. She and her son were discharged home.

#### 3.2.5. Evaluation—So What?

The return to normal life was not as easy or smooth as claimed due to patient’s concerns and anxiety about the possibility of reinfection with MERS-CoV.

“I was happy to see people but at the same time, I was afraid that I would be exposed to MERS-CoV, though I know I have antibodies. They told me that even if you get exposed to MERS-CoV patients, you won’t get it. But until now, I still have this fear. Once I enter the ICU, I feel smothered.”

At the same time, she was tormented not only by psychological suffering, but also by physical limitations due to neuropathy. The patient explained that it was a lonely journey, that she found it difficult to explain to people what she had been through and that she needed time to return back to normal. Even her husband did not want to accept her temporary restrictions. This, in turn, created inner anger towards the people around her.

“…during one of my visits to the neuroscience clinic, the doctor asked me if I have angry feelings, I replied I feel angry toward my husband, and he replied why, I said: I don’t know but anything he says gets me angry, then he said: you need to visit a psychiatrist, I said: no I don’t need [crying] he again said: yes you need this. At that time I cried. He said: you have to go…”

The patient was referred to a psychiatrist, with whom she shared her concerns. In fact, when asked by the psychiatrist whether or not the patient had suicidal thoughts, the patient responded “no, never.” She claimed that she wanted to live to take care of her son but feared that she would not pay enough attention to him. After chronic anxiety, anger and severe insomnia, she was diagnosed with post-intensive care syndrome (PICS). Thus, she started taking antidepressants in 2016, 6 months before her husband’s death, and continued taking them thereafter. While the patient was trying to normalize and stabilize her life, she was in a constant state of anxiety that she might not be able to breathe or would run out of air at any moment. While some of these fears have subsided over time, many of them still haunt her. Moreover, these concerns have increased dramatically during the COVID-19 outbreak in 2020.

“During the Corona pandemic, I didn’t want to be admitted to ICU and intubated. I didn’t want anything to happen to my family. I didn’t want them to go through the same experience before. When I had flu symptoms, I wore a mask at home, so no one would have Corona. I don’t know. I am afraid that maybe I have Corona. Maybe I am the first one in my family and our department in the hospital, once they allowed the vaccine, I went and took it. Everybody was against that; they did not want to be in an experiment and receive the vaccine at the beginning and so on. To me, I didn’t care. I went through it, and I took the vaccine.”

These fears also affected her as a mother on a physical and psychological level.

“All the time, I wake up and check on him if he is breathing or not. I was afraid that maybe someone would take him away from us, or maybe someone would excruciate him, or maybe I will die. All the time I had this fear... I did not want him to be out of my sight, I always want to take care of him.”

Alas, the enormous care and concern for her son was overshadowed by physical limitations that the patient currently cannot explain to the baby, who requires constant attention and play.

“He always asks me, why you are always tired? why you can’t do a lot of activities? Sometimes he wants me to run with him and jump, so I do for a while then I feel tired. I tell him I am sorry I am tired, and he asks why. I explain to him to let him understand [tears].”

Additionally, the psychological and physical impact of this experience motivated her to make a career decision. She decided to change the nature of her job from an ICU nurse to a quality nurse because she was not ready to work in the ICU again. The patient applied for a promotion and was given a different position. Although she no longer goes to the intensive care unit and sometimes misses this position, the fear is still there, and she does not want to return to it.

“Ummm, there is fear until now. Once I enter the ICU, I feel I don’t want to go through the same experience.”

#### 3.2.6. Coda—How Does It All End?

Despite the enormous amount of suffering and difficulty that the NDE entailed, the patient emphasized the positive impact it had on her life. She also highlighted the changes for the better and the decisions she has made on a personal and professional level. She has become more open to life and more respectful of her family and the precious time she spends with them.

“I got excited and enrolled myself in the driving school. I also traveled as I wanted to enjoy my life more than staying at home, as I wanted to use every moment, I enjoy it. I want to travel, I want to experience new things, I want to get to know more people, and talk to people more. I want to learn everything, so I involve myself in training courses, workshops, and conferences, I tried as much as I can. Nowadays, I reduced my activities, I no longer go outside a lot, I enjoy peace more, I want to spend more time with my family, and I don’t want to go outside a lot.”

The patient also felt a moral obligation to unconscious patients who have experienced NDEs, and her unique experience has allowed her to walk in their shoes. She devoted most of her educational efforts to these patients.

“I love to share my story, to be honest, especially my experience as a patient. What do patients feel? do they hear people when they talk? when they are sedated? How it is very important to listen to your patient. When I told them I cannot stand, they were telling me No stand-up. I know my capabilities; I know that I cannot set them. They were forcing me to sit and once I fell, they held me quickly before I reached the floor. I was telling them “I feel severe pain as if I am falling from the 10th floor. No one understood the point (feeling) because I had severe neuropathy and muscle waste. I developed nerve damage, but no one noticed it. There are things like that. The hallucination that I have. I bring it in a funny way, but at the same time, if I did not share it at that time, no one would have known that I went through hallucinations when I was in ICU.”

At present, the patient strives to apply the positive lessons she has learned to her son’s life by making him open to life and a lover of trying various things.

“Currently, I let him expose himself to life, see more people, and enjoy life. I think about his health, even more, I want him to have community and have more friends. He is a shy person, very shy. He used to have only one friend, but now thanks to God, has started to socialize more.”

## 4. Discussion

This study depends on narrative analysis to adhere to key characteristics that enhance both the validity and accessibility of the study. These include a thorough investigation of structure, characterization, setting, language, theme exploration, and nurse engagement. By methodically addressing these sections, the study guarantees a complete understanding of the story, enabling readability for readers.

According to prevailing views in neuroscience, NDEs occurring after severe head trauma, critical illness, or coma are often life-transforming experiences in which no awareness or sensory experience of any kind is possible [19,23]. The causes of NDEs are complex and not fully understood, and while many medical and psychological rationales have been proposed, they remain hypothetical and often fail to explain the entire phenomenon. Although there are general patterns, each case is quite different from the other and requires accurate recording and reporting to potentially explain the phenomenon [7,23,24].

This narrative study examined a pregnant woman’s NDE due to complications from MERS-CoV. It was an experience filled with emotional and physical stress and pressure. Although most experiences recounted by patients are considered subjective, as in the case of this patient, certain developments indicate that these experiences are valid. According to Sabom (1982), NDEs are divided into three types: out-of-body; transcendental, including the transition of consciousness to another dimension; and a combined experience [25]. In the case of this study, the patient reported that she experienced all three types of near-death experiences, finding herself in the desert or a white palace, all the while searching for her son. This suggests that during the NDE, the patient was somewhat conscious and, even while moving from one dimension to another, did not leave thoughts about her son, who was born while the mother was unconscious in the intensive care unit.

Additional evidence suggests that people who have had NDE heard conversations and observed the actions of people around them during their comatose state. This pattern was also true for the patient in this study, who remembered all the conversations that took place at her bedside and even remembered the people who said certain things [26,27]. An earlier study by Woollacott and Peyton (2021) reported similar findings in a pregnant woman who had an NDE in the seventh month of pregnancy and recalled all the details of medical discussions about complications she faced [28]. Similarly, an earlier study by Purkayastha and Mukherjee (2012), which reported three cases of near-death experiences in different age groups, found that all three patients stated that they initially remembered events that occurred during that time. However, after some time, all three patients could not accurately remember the events that had occurred [24]. In contrast, the patient in this study remembered all the details, with some confusion of details.

In addition to real-time memories of the NDE, the condition itself leaves lasting consequences, as shown in this study and confirmed by earlier studies [25,26,27,28]. Although some of these effects may not be related to the NDE itself, people often associate them with the NDE. In this case, the patient reported a constant fear of suffocation and physical consequences such as neuropathy, which required her to recover for quite a long time, as well as leading to many years of insomnia and a fear of her son’s death rather than that of herself. Although a discussion of the vast number of such cases is beyond the scope of this article, it is noteworthy that an NDE often lasts just a few minutes, but usually leads to significant and lifelong transformations of beliefs and values [4,5,19].

An emerging pattern observed in this study was that the patient went through the same traumatic experiences in terms of memories during COVID-19, which became her main incentive to receive the vaccines to protect her family and son so that they would not experience the same pain. This shows that suffering does not end with their life signs improving in MERS-CoV or COVID-19, and the effects may be immediate or may develop over time. In this case, the patient experienced significant changes and difficulties for 8 years after contracting MERS-CoV. Her experience has shown that these patients require a long recovery period with extensive support from others. This support should consist of support at a professional level, especially for health care providers. In this case, the patient asked to be transferred from intensive care to another department and was supported by management [24,29].

Despite these negative consequences, the patient also experienced what is called an NDE aftereffect, that is, changes in beliefs and values [30,31,32]. Thus, consistent with previous evidence, it has been shown that a decreased fear of death may be a long-term consequence of near-death experiences. She began to lead a more altruistic life, and sought an interest in the meaning of life, while her materialistic values decreased and her interest in traveling and exploring the world increased. At the same time, the patient reconsidered the value of her family and those who supported her during the long process of recovery. The general experience of the Saudi patient, supported by previous cases, suggests that although NDE survivors usually recover over time, certain fears still remain with them throughout life that are not necessarily related to the NDE, but will always have a subtle connection [7,30,31,32].

This study depends on a framework that thoroughly examines various aspects. It is essential to highlight that this study used two consecutive interviews to ensure an in-depth investigation of the situation. Furthermore, the rich and detailed data provided by the narrative design, humanizing approach, holistic understanding, flexibility in the interview and data collection, and interpretive insights can lead to policy implications, while narrative analysis can provide insights that can inform policy decisions and interventions, particularly in areas such as health, social policy, and development. It is also important to mention that this study has a limitation due to the use of the Greyson scale instead of the NDE-C scale; this is because the data collection preceded the NDE-C scale’s development, and therefore, the researchers were not able to include it in the current study. This study also recognizes the impact of having small sample size and the narrative’s temporal context, which limits generalizability. Additionally, this study used a subjective approach, which may have limited the accuracy of the collected data and the control over data collection, in addition to the constraint of it being time-consuming. Furthermore, it is crucial to mention that the study’s lack of predictive power is a significant limitation. The use of narrative analyses focuses on retrospective data rather than future events. This study does not imply anticipating future developments.

## 5. Conclusions

In conclusion, this study used a Greyson scale to investigate an NDE with a total score of 12. The patient experienced different kinds of NDEs, including out-of-body experiences and moments of transcendence. She also dealt with persistent hallucinations, neuropathy, and post-intensive care syndrome. Remarkably, the NDE sparked aftereffects, leading to a change in the patient’s beliefs and values.

NDE survivors should be encouraged to talk more and share their stories with others if they wish. This can help them heal from the inside, and their stories can help others heal, not just after an NDE, but in everyday life. In addition to its scientific value, this approach may also have significant human value; listening to these experiences can give people some comfort and a moment of reflection. This study not only investigates NDEs but also considerably adds to the existing literature in various ways, firstly by integrating a varied range of cultural views, particularly because it is a case that is situated outside of the US and other Western nations, which tend to dominate NDE literature. Moreover, our study highlights the essential role of healthcare providers in the consequences of NDEs; the study shows the importance of communication with comatose patients. Furthermore, this study underscores the requirement of compassion when reacting to patients who have NDEs. This highlights the need to improve the training and awareness of health providers regarding the care of patients with NDEs.

## Figures and Tables

**Table 1 healthcare-12-00298-t001:** Labovian narrative framework adapted by Simpson (2004, p. 115) [21].

Narrative Category	Narrative Question	Narrative Function	Linguistic Form
Abstract	What was this about?	Signals the beginning, draws attention from the listener	A short summarizing statement, provided before the narrative begins
Orientation	Who or what was involved in the story, and when and where did it take place?	Identifies time, place, characters, activities and situation of the story	Past continuous verbs, adverbials of time, place, and manner
Complicating action	Then, what happened?	Core narrative that provides the “what happened?” element	Temporally ordered narrative clauses with a verb in the present or simple past
Resolution	What finally happened?	Tells the final key event of the story	The last of the narrative clauses that began the Action
Evaluation	So what?	Shows how the story is interesting, stresses the most peculiar elements	Intensifiers, modal verbs, repetition, embedded speech, and evaluative commentary
Coda	How does it all end?	Signals the end, links back to present situation	A generalized statement

**Table 2 healthcare-12-00298-t002:** Patient’s Greyson scale results.

Question	Answer/Score
**1. Did time seem to speed up or slow down?** 0 = No 1 = Time seemed to go faster or slower than usual 2 = Everything seemed to be happening at once; or time stopped or lost all meaning	2
**2. Were your thoughts speeded up?** 0 = No 1 = Faster than usual 2 = Incredibly fast	1
**3. Did scenes from your past come back to you?** 0 = No 1 = I remembered many past events 2 = My past flashed before me, out of my control	0
**4. Did you suddenly seem to understand everything?** 0 = No 1 = Everything about myself or others 2 = Everything about the universe	0
**5. Did you have a feeling of peace or pleasantness?** 0 = No 1 = Relief or calmness 2 = Incredible peace or pleasantness	0
**6. Did you have a feeling of joy?** 0 = No 1 = Happiness 2 = Incredible joy	0
**7. Did you feel a sense of harmony or unity with the universe?** 0 = No 1 = I felt no longer in conflict with nature 2 = I felt united or one with the world	2
**8. Did you see, or feel surrounded by, a brilliant light?** 0 = No 1 = An unusually bright light 2 = A light clearly of mystical or other-worldly origin	2
**9. Were your senses more vivid than usual?** 0 = No 1 = More vivid than usual 2 = Incredibly more vivid	2
**10. Did you seem to be aware of things going on elsewhere, as if by extrasensory perception (ESP)?** 0 = No 1 = Yes, but the facts have not been checked out 2 = Yes, and the facts have been checked out	0
**11. Did scenes from the future come to you?** 0 = No 1 = Scenes from my personal future 2 = Scenes from the world’s future	0
**12. Did you feel separated from your body?** 0 = No 1 = I lost awareness of my body 2 = I clearly left my body and existed outside it	2
**13. Did you seem to enter some other, unearthly world?** 0 = No 1 = Some unfamiliar and strange place 2 = A clearly mystical or unearthly realm	1
**14. Did you seem to encounter a mystical being or presence, or hear an unidentifiable voice?** 0 = No 1 = I heard a voice I could not identify 2 = I encountered a definite being, or a voice clearly of mystical or unearthly origin	0
**15. Did you see the deceased or religious spirits?** 0 = No 1 = I sensed their presence 2 = I actually saw them	0
**16. Did you come to a border or point of no return?** 0 = No 1 = I came to a definite conscious decision to “return” to life 2 = I came to a barrier that I was not permitted to cross; or was “sent back” against my will.	0
**Total**	**12**

## Data Availability

Data can be requested from authors.

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
