# Peer review of "Nurse Who Had MERS-CoV Complications with A Near-Death Experience during Pregnancy: A Narrative Analysis"

_healthcare, 2024, doi:10.3390/healthcare12030298_

Round 1
Reviewer 1 Report
Comments and Suggestions for Authors
Review for Healthcare
Nurse who had MERS-CoV complications with near-death experience during pregnancy: A narrative analysis
Overall this is an interesting manuscript that I think makes a very useful contribution to the literature, particularly because it is a case that is situated outside of the US and other Western nations – which tend to dominate NDE literature. So, for that reason alone, it should be published as it adds cultural diversity to studies in this area.
Equally, this case is interesting as it also discusses the less positive transformations that occur after NDE, along positive ones. Overwhelmingly, NDE literature speaks most often to positive transformation, so it is good to see more balance in the aftereffects, even if they might reasonably be attributed to the MERS-CoV.
However, there are a number of issues that I think would be useful to address prior to acceptance of the manuscript for publication:
1. It would be useful to mention that the Greyson scale, while unprecedented in its contribution to NDE research, has recently been critiqued and a new scale (the NDE-C scale) has been created. While I personally do not think the NDE-C scale is an improvement, it would simply be useful for the authors to note they are aware of it but did not use it because their data collection period occurred before the NDE-C scale was published.
2. When mentioning NDE, there is no need to write “NDE experience”. This does not make sense as the ‘E’ in NDE refers to experience. There are a few times throughout the manuscript where this has occurred.
3. It would be useful to strengthen the conclusion. Specifically, it would be good to state some ways that this manuscript contributes to the overall literature. I have provided one example above of how it does; that is, it adds cultural diversity to the existing NDE literature base. Another point that may be useful to more strongly mention as a contribution (although it is briefly touched on already), is that the study supports the need for health professionals to both be aware of what they are saying while a patient is comatose, as well as be aware of how they respond to patients who may have had an NDE – suggesting they go to a psychiatrist is usually not helpful!
Comments on the Quality of English LanguageCould be improved for minor instances where there is lack of clarity.
Author Response
Dear reviewer
I am writing to submit the revised manuscript; we did all required corrections which certainly improved the manuscript.
- It would be useful to mention that the Greyson scale, while unprecedented in its contribution to NDE research, has recently been critiqued and a new scale (the NDE-C scale) has been created. While I personally do not think the NDE-C scale is an improvement, it would simply be useful for the authors to note they are aware of it but did not use it because their data collection period occurred before the NDE-C scale was published.
Response: We appreciate the feedback and acknowledge the existence of the NDE-C scale in recent NDE research. While the Greyson scale has indeed played a crucial role in our study and has been widely recognized for its contributions to Near-Death Experience (NDE) research, we are aware of the NDE-C scale. It's important to note that our data collection period predates the publication of the NDE-C scale, and therefore, we were not able to incorporate it into our study.
- When mentioning NDE, there is no need to write “NDE experience”. This does not make sense as the ‘E’ in NDE refers to experience. There are a few times throughout the manuscript where this has occurred.
Response: We appreciate your feedback, we considered and corrected that.
- It would be useful to strengthen the conclusion. Specifically, it would be good to state some ways that this manuscript contributes to the overall literature. I have provided one example above of how it does; that is, it adds cultural diversity to the existing NDE literature base. Another point that may be useful to more strongly mention as a contribution (although it is briefly touched on already), is that the study supports the need for health professionals to both be aware of what they are saying while a patient is comatose, as well as be aware of how they respond to patients who may have had an NDE – suggesting they go to a psychiatrist is usually not helpful!
Response: Thank you for your comment, we added paragraph to strengthen the conclusion.
Yours Faithfully

Reviewer 2 Report
Comments and Suggestions for Authors
The authors investigate a NDE using a narative analysis type study. Notably, a great deal of effort was put into organizing the manuscript. However, there are some issues.
Abstract:
-The conclusions should be strengthened and made explicit.
Introduction:
-Sentences 48-49 should be rewritten
-Line 57 from personal experience?
-Line 77-78, can you expand it in a few phrases
Materials and Methods
-description of methods should be expanded ( here or in the discussion section)
-Line 122-123, can you expand into a few phrases
-Reliability and validity:OK
-Procedures:OK
3. Results
-Descriptive data: OK
-Qualitative data: OK
Discussions
-The characteristics of narrative analyses should be stated and explained in a few lines (for the manuscript to gain validity and be more accessible to readers)
Furthermore, advantages and study limitations should be presented like in any study.
Only as a suggestion- advantages: Rich and detailed data:, Humanizing approach, Holistic understanding:, Flexibility:, Interpretive insights, Appropriate for sensitive topics, Can lead to policy implications(Narrative analysis can provide insights that can inform policy decisions and interventions, particularly in areas such as health, social policy…. )
Only as a suggrestion- drawbacks( limitations): Subjectivity, Limited generalizability, Ethical considerations:, Limited control over data collection, Time-consuming, Interpretation challenges:, Limited statistical analysis ……
Lack of the study's predictive power should also be stated
Conclusions
The conclusion should be revised to reflect the results.
Author Response
Dear reviewer
I am writing to submit the revised manuscript; we did all required corrections which certainly improved the manuscript.
Abstract:
-The conclusions should be strengthened and made explicit.
Response: The conclusion of abstract was strengthened.
Introduction:
-Sentences 48-49 should be rewritten
Response: rewritten
-Line 57 from personal experience?
Response: Revised
-Line 77-78, can you expand it in a few phrases
Response: expanded
Materials and Methods
-description of methods should be expanded ( here or in the discussion section)
Response: we expanded the description of methods in methods section
-Line 122-123, can you expand into a few phrases
Response: Expanded
-Reliability and validity: OK
Response: Thank you
-Procedures: OK
Response: Thank you
- Results
-Descriptive data: OK
Response: Thank you
-Qualitative data: OK
Response: Thank you
Discussions
-The characteristics of narrative analyses should be stated and explained in a few lines (for the manuscript to gain validity and be more accessible to readers)
Response: added
Furthermore, advantages and study limitations should be presented like in any study.
Response: added
Only as a suggestion- advantages: Rich and detailed data:, Humanizing approach, Holistic understanding:, Flexibility:, Interpretive insights, Appropriate for sensitive topics, Can lead to policy implications(Narrative analysis can provide insights that can inform policy decisions and interventions, particularly in areas such as health, social policy…. )
Only as a suggrestion- drawbacks( limitations): Subjectivity, Limited generalizability, Ethical considerations:, Limited control over data collection, Time-consuming, Interpretation challenges:, Limited statistical analysis ……
Response: added
Lack of the study's predictive power should also be stated
Response: added
Conclusions
The conclusion should be revised to reflect the results.
Response: revised
Yours Faithfully

Reviewer 3 Report
Comments and Suggestions for Authors
Regardless of prior published solid neurobiological explanations, near-death experiences become part of survivors' presence and meaning in the world. Therefore, healthcare professionals and loved ones must recognize the experience of NDE and support individuals in their journey of constructing meaning. It is noteworthy that other patients with NDEs felt stigmatized. Educating the public, healthcare professionals and the family on how to respond, support, and relate to survivors with NDEs is important in their journey of life transformation.
The features of NDE – out-of-body experience, entering a transitional tunnel, encountering a bright light and finally emerging into it – described by participants are similar to previous reports. Anyhow, each experience is different; thus, each patient will require a personalized supporting environment. The evidence is mixed as to whether prevailing social conventions, cultural experiences, and religious notions influence NDE reports. This narration revealed that the patient found herself thrown into a reality with a new set of limitations and new possibilities in every aspect of her life. As other cases reported, such experiences drew the participants' attention to their mortality but also caused them to fear a recurrent episode. The story illustrates the importance of throwness as the point where they realize they are confronted with a new beginning with their world radically transformed. However, participants' loved ones are also thrown into a new reality without going through a similar process of transformation. In this case, we have another trigger – the newborn. It is the reason for a great fight and subsequent transformation. The difficulties in recovery are understandable after a long period of lying flat on a bed.
I have a few issues to be addressed by the authors.
Point 1 Lines 67 – 76 comprise too much information on MERS-Cov infection. It is not directly the subject of this article; thus, it is redundant.
Lines 173-174 After 32 days of labor should be rephrased, as I think that the authors mean 32 days after birth (c-section in this particular case, no labor involved)
Line 188 The word patient is wrote incorrect
Comments on the Quality of English Language
The English language is fine
Author Response
Dear reviewer
I am writing to submit the revised manuscript; we did all required corrections which certainly improved the manuscript.
Regardless of prior published solid neurobiological explanations, near-death experiences become part of survivors' presence and meaning in the world. Therefore, healthcare professionals and loved ones must recognize the experience of NDE and support individuals in their journey of constructing meaning. It is noteworthy that other patients with NDEs felt stigmatized. Educating the public, healthcare professionals and the family on how to respond, support, and relate to survivors with NDEs is important in their journey of life transformation.
Response: thank you for your comment which highlighting the importance of this data.
The features of NDE – out-of-body experience, entering a transitional tunnel, encountering a bright light and finally emerging into it – described by participants are similar to previous reports. Anyhow, each experience is different; thus, each patient will require a personalized supporting environment. The evidence is mixed as to whether prevailing social conventions, cultural experiences, and religious notions influence NDE reports. This narration revealed that the patient found herself thrown into a reality with a new set of limitations and new possibilities in every aspect of her life. As other cases reported, such experiences drew the participants' attention to their mortality but also caused them to fear a recurrent episode. The story illustrates the importance of throwness as the point where they realize they are confronted with a new beginning with their world radically transformed. However, participants' loved ones are also thrown into a new reality without going through a similar process of transformation. In this case, we have another trigger – the newborn. It is the reason for a great fight and subsequent transformation. The difficulties in recovery are understandable after a long period of lying flat on a bed.
Response: thank you for your comment which highlighting the importance of this data.
I have a few issues to be addressed by the authors.
Point 1 Lines 67 – 76 comprise too much information on MERS-Cov infection. It is not directly the subject of this article; thus, it is redundant.
Response: thank you for your comment, we revised this part.
Lines 173-174 After 32 days of labor should be rephrased, as I think that the authors mean 32 days after birth (c-section in this particular case, no labor involved)
Response: We appreciate your comment, we revised this part
Line 188 The word patient is wrote incorrect
Response: corrected
Yours Faithfully

Round 2
Reviewer 2 Report
Comments and Suggestions for Authors
The author addressed all the issues.